# Synthesis and Characterization of a Self-Polycondensation Diazaphthalanone Monomer and Its Polymers from Polycondensation Reactions

**DOI:** 10.3390/polym14183904

**Published:** 2022-09-19

**Authors:** Xin Liu, Xiaozhou Zhang, Jiawei Jiang, Hongge Jia, Xigao Jian, Jinyan Wang

**Affiliations:** 1College of Materials Science and Engineering, Qiqihar University, Wenhua Street 42, Qiqihar 161006, China; 2Heilongjiang Province Key Laboratory of Polymeric Composition Material, Qiqihar 161006, China; 3Department of Polymer Science & Engineering, Dalian University of Technology, Dalian 116024, China

**Keywords:** poly(aryl ether ketone), solubility, thermal properties

## Abstract

Polyether ketone (PEK) plastics are linear thermoplastic polymers connected by at least one ether bond and at least one ketone bond on the aryl group. The reason for their excellent heat resistance, rigidity, and mechanical strength is that their main molecular chain contains plenty of aromatic rings and polar carbonyl groups, and their molecular chain presents a large rigidity and strong intermolecular force. In addition, the main chain contains a considerable number of ether bonds, resulting in a certain toughness. However, polyether ketone materials have the disadvantage of poor solubility because of their excellent rigidity. To improve the solubility of polyether ketone, the preparation method of a novel nitrogenous heterocyclic polyaromatic ether monomer, 2-(4-chlorophenyl)-2,3-dihydrophthalazine-1,4-dione (CDD), was proposed, and its activity of polymerization was studied. The average molecular weight of the poly(aryl ether ketone) containing a nitrogenous heterocyclic polyaromatic ether group obtained by self-polycondensation of CDD was 4.181 × 10^3^ kg/mol, and the yield was 90.5%. In order to further explore the activity of monomers, novel copolymerized poly(aryl ether ketone) (PBCD) containing a nitrogenous heterocyclic polyaromatic ether structure was prepared by ternary copolymerization with 4,4-difluorobenzophenone (DFBP) and bisphenol fluorene (BHPF) with high activity. The average molecular weight of PBCD was 72.793 × 10^3^ kg/mol, the molecular weight distribution was 2.344, and the yield was 88.1%. Fourier transform infrared spectroscopy (FT-IR) and ^1^H NMR were used to confirm the structure of the obtained polymer. Through thermogravimetric analysis (TGA), the determined weight loss temperature of 5% under nitrogen was higher than 500 °C, indicating excellent thermal stability. Compared with the solubility of the binary copolymer containing fluorenyl poly(aryl ether ketone) (PBD), the polymer showed reasonable solubility in selective solvents such as chloroform and *N*,*N*-dimethylacetamide.

## 1. Introduction

In recent years, high-performance engineering plastics and polyether ketone materials have been widely used in many emerging fields of science and technology. The main chain of the polymer is mainly composed of a benzene ring and ether bond, creating polyether ketone materials with high strength, heat resistance, size stability, and other excellent properties [1,2,3]. These materials are valued and applied in the fields of electronics and electrical appliances, machinery, aerospace, etc. For example, they are mainly used in the electronics industry for wire, magnetic conductor cladding, high-temperature terminal posts, terminal boards, and scram wire printed circuit boards. However, due to the low content or lack of flexible aliphatic groups in the main chain, their solubility is poor, and their application in some special fields is limited [4,5]. They are also difficult to process and can only be processed by thermal forming, while other processing methods, such as blow molding, are also difficult to achieve [6].

Nitrogenous heterocyclic poly(aryl ether ketone) is a new kind of special performance heterocyclic polymer, and its excellent performance is attributed to the rigid and asymmetric phenyl and nitrogen-containing groups in the main chain of the polymerization [7]. Asymmetric phenyl and nitrogen-containing groups show non-planar structures to each other in spatial configuration so that the polymer retains rigidity and increases the characteristics of its non-planar structure [8], and the gap and free volume of the main chain become larger. Since Hay et al. [9] synthesized the first nitrogen-containing polymer by N-C coupling reaction using 2-dihydro-4-(4-hydroxyphenyl)-1 (2H)-phthalazone and activation of the halogen compound as monomers in 1993, more and more attention has been paid to nitrogen-containing polymers. Starting with the molecular structure of monomers, many structural polymers containing nitrogen have been synthesized with excellent properties, such as poly (phthalazone) ether [10,11,12,13,14,15,16].

With the development of science and technology, the high-tech field has put forward higher requirements for materials, such as molecular weight [17], glass temperature [18], heat resistance [19], solubility [20], mechanical properties [21], and other aspects for the requirements of higher performance. The aromatic heterocyclic structure containing nitrogen atoms has a conjugated system [22], which not only retains the overall rigidity of the main chain but also inhibits the free rotation of chain segments, endowing the polymer with excellent thermal and mechanical properties. It also increases the free volume of the backbone so that the solvent can more easily enter the backbone and improve the solubility of the polymer [23].

Poly(aryl ether ketone) resin is usually made of bisphenol monomer and active bishalogen monomer as the raw materials [8,24,25] (Aa, Bb double monomer polycondensation) under the catalysis of potassium carbonate, known as solution polycondensation [26,27]. This kind of self-polymerizing monomer contains two functional groups that can react with each other, so the influence of the number of groups of two monomers on the degree of polymerization can be avoided during polymerization [28], and theoretically, high molecular weight polymers can be obtained. Therefore, this study intends to design and synthesize two new autocondensation monomers containing a phthalazone structure from the perspective of molecular structure design. The synthesized monomers will contain both phenolic hydroxyl and halogen groups. Through the self-polymerization reaction of the new monomers, the polymerization properties of the monomers will be studied, providing new ideas for the synthesis of poly(aryl ether ketone) resins.

## 2. Materials and Methods

### 2.1. Materials

Phthalic anhydride (PA, 99%, Innochem Science & Technology Co., Ltd., Beijing, China) and 4-chlorophenylhydrazine hydrochloride (CPH, 97%, Innochem Science & Technology Co., Ltd., Beijing, China) were used as received without further purification. 4,4′-(9H-Fluorene-9,9-diyl) diphenol (BHPF, Aladdin, 98%, Aladdin Biochemical Technology Co., Ltd., Shanghai, China) and 4,4′-Difluorobenzophenone (DFBP, Aladdin, 99%, Aladdin Biochemical Technology Co., Ltd., Shanghai, China) were used as received without further purification. All other chemicals were purchased from Tianjin Comeo Chemical Reagent Co., Ltd. (Tianjin, China) and used as received.

### 2.2. Characterization Methods

The Fourier transform infrared (FT-IR) spectrum was recorded by reflection method with a Thermos Nicolet Nexus 470 FT-IR spectrometer (Thermo Fisher Scientific, Waltham, MA, USA). The ^1^H NMR spectrum was measured on a Bruker AVANCE-600 MHz instrument (Bruker, Billerica, MA, USA) with chloroform/dimethyl sulfoxide (CDCl_3_/DMSO) as the solvent and tetramethyl silane (TMS) as an internal standard. Thermal gravimetric analysis (TGA) of the polymer sample was measured on a Q5000IRS (TA Instruments, New Castle, DE, USA) instrument at a heating rate of 10 °C/min in N_2_. Gel permeation chromatography (GPC) analyses were carried out on a PL-GPC 120 (Agilent Technologies, Santa Clara, CA, USA) instrument that used tetrahydrofuran as a solvent to test molecular weight and molecular weight distribution.

### 2.3. Monomer Synthesis

#### Synthesis of Novel Nitrogenous Heterocyclic Polyaromatic Ether Monomer CDD

A mixture of PA (10 mmol) and CPH (10 mmol) was dissolved in a solvent with a certain volume ratio of DMF and AcOH and heated by microwave in a microwave reactor under a certain power and reaction time under a nitrogen atmosphere. The reaction was then terminated, and the product was cooled to room temperature, poured into cold water for recrystallization, washed with water, and suction filtered to obtain yellow needle-like crystals, which were then dried. The yellow needle-like crystals of the product were CDD, and the yield of the product was 69.9% (Figure 1). Selected data of CDD: ^1^H-NMR (600 MHz, DMSO, ppm): δ = 6.78–6.80 (dd, J = 10.2, 1.8 Hz, 2H, Ar-H-N), 7.19–7.20 (d, J = 6.0 Hz, 2H, Ar-H-Cl), 7.92–7.96 (m, J = 19.8, 9.8, 4.8 Hz, 4H, Ar-H), and 8.71 (s, ^1^H, -NH); FTIR (KBr): 3360 (-NH), 3100, 3030 (Ar-H), 1708 (C=O), 1592, 1490 (Benzene ring skeleton), and 870 (Ar-Cl) cm^−1^.

### 2.4. Polymer Synthesis

#### 2.4.1. Self-Polycondensation of CDD-Synthesis of PCDD

The synthesis method of CDD-K is shown in Figure 2. Potassium hydroxide (2 mmol) and isopropanol (25 mL) were added to a three-necked flask under the protection of nitrogen. Then, CDD (2 mmol) was slowly added and heated to 88 °C under reflux for 2 h after all KOH was dissolved. Then, the mixture was washed with isopropanol and filtered to obtain a white powder, 3-(4-chlorophenyl)-4-oxo-3,4-dihydrophthalazine-1-ester potassium (CDD-K), vacuum-dried for 24 h after heating ceased, and cooled to room temperature. Additionally, the yield of the product was 79.06%. FTIR (KBr): 3355 (-OK) and 1636 (C=O, C=N) cm^−1^.

The specific synthetic route of CDD self-condensation PCDD is shown in Figure 2. Under nitrogen protection, a mixture of CDD-K (2.58 mmol) and K_2_CO_3_ (3.87 mmol) was added to the three-necked flask equipped with a water separator, and 10 mL of tetramethylene sulfone were added before stirring. Toluene was used as the water separating agent, the reaction temperature was 200 °C for 10 h, and the reaction ended when the viscosity of the system no longer increased. The reaction solution was poured into hot water, and the polymer was precipitated in reverse. After purification, the yield was calculated by drying and weighing, and the yield was 90.5%. FTIR (KBr): 1680 (C=O) and 1256 (C-O-C) cm^−1^.

#### 2.4.2. Synthesis of Ternary Copolymerized Poly(aryl ether ketone)-Synthesis of PCBD

The specific synthesis route of CDD ternary copolymerization PCBD is shown in Figure 3. Under nitrogen protection, CDD (1.50 mmol), bisphenol fluorene (BHPF) (1.50 mmol), 4,4-difluorobenzophenone (DFBP) (1.50 mmol), and K_2_CO_3_ (2.25 mmol) were placed in a solution containing water. In the three-necked flask of the device, DMAc (15 mL) was added as the reaction solvent began to stir. Toluene was used as the water separating agent, and the ternary copolymerization reaction was carried out at 160 °C for 10 h until the viscosity of the reaction solution no longer increased. Then, the hot reaction solution was poured into a mixture of methanol and water with a volume ratio of 1:1, and reverse precipitation was carried out on the polymer. After purification, it was dried and weighed, and the yield was 88.1%. ^1^H-NMR (600 MHz, CDCl_3_, ppm): δ = 6.93–6.95 (t, J = 6.0 Hz, 4H, Ar-H), 6.99–7.02 (d, J = 18.0 Hz, 4H, Ar-H), 7.14 (s, 2H, Ar-H), 7.22–7.31 (t, J = 5.4 Hz, 6H, Ar-H), 7.38–7.42 (dd, J = 18.5, 13.9 Hz, 6H, Ar-H), and 7.74–7.79 (t, J = 15.0 Hz, 10H, Ar-H). FTIR (KBr): 1650 (C=O), 1245 (C-O-C), and 745 (cardon ring) cm^−1^.

#### 2.4.3. Synthesis of Binary Copolymerized Poly(aryl ether ketone)

The specific synthetic route of binary copolymerization PBD is shown in Figure 4. The synthetic route is the same as PCBD and PBOD. BHPF (6 mmol), DFBP (6 mmol), and K_2_CO_3_ (9 mmol) were added to a three-necked flask equipped with a water separator under nitrogen protection. DMAc was used as the reaction solvent, and toluene was used as the water-carrying agent. The mixture was stirred at 160 °C for 6 h until the viscosity of the reaction solution no longer increased and stopped heating. After the reaction was complete, it was poured it into the water while it was still hot and the polymer was reversely precipitated. The yield was 93.8%. ^1^H-NMR (600 MHz, DMSO, ppm): δ = 6.85–6.87 (d, J = 12.0 Hz, 4H, Ar-H), 7.13–7.15 (d, J = 12.0 Hz, 4H, Ar-H), 7.29–7.39 (dt, J = 30.0, 1.8 Hz, 10H, Ar-H), 7.64–7.72 (d, J = 48.0 Hz, 4H, Ar-H), and 8.11 (s, 2H, Ar-H). FTIR (KBr):1655 (C=O), 1245 (C-O-C) and 745 (cardon ring) cm^−1^.

## 3. Results and Discussion

### 3.1. Structural Characterization of Novel Monomer CDD

The obtained target product CDD was tested by infrared spectroscopy and the results of CDD infrared spectroscopy are shown in Figure 1.

As can be seen from Figure 1, 3360 cm^−1^ corresponds to the -NH stretching vibration peak, 3100 cm^−1^ and 3030 cm^−1^ correspond to the Ar-H stretching vibration peak, and 1708 cm^−1^ corresponds to the C=O stretching vibration peak. The stretching vibration peaks of the benzene ring skeleton at 1592 cm^−1^ and 1490 cm^−1^ correspond to the stretching vibration peaks of Ar-Cl at 870 cm^−1^. According to the analysis, the functional groups of the target product are reflected in the infrared spectrum.

The hydrogen NMR spectrum of target monomer CDD is shown in Figure 2.

Figure 2 shows that in the ^1^H-NMR spectrum, the chemical shift is 6.78–6.80 ppm, corresponding to the absorption peak of the number 2 Ar-H-N hydrogen atom in the chemical formula of the product, which is a double peak with an integral area of 1.00. The chemical shift is 7.19–7.20 ppm, corresponding to the absorption peak of the number 3 Ar-H hydrogen atom in the chemical formula of the product, which is a double peak with an integral area of 1.00. The chemical shift is 7.92–7.96 ppm, corresponding to the absorption peak of the number 1 Ar-H hydrogen atom in the chemical formula of the product, which contains multiple peaks with an integral area of 2.01. The chemical shift is 8.71 ppm, corresponding to the absorption peak of the number 4 -NH hydrogen atom in the product chemical structure formula, which is a single peak with an integral area of 0.50. The hydrogen atoms in the molecular structure of CDD can be ascribed to the hydrogen spectrum of nuclear magnetic resonance, so the product was successfully synthesized. The product was identified as a novel phthalazone monomer CDD by IR and NMR.

### 3.2. Study on Self-Condensation Polymerization of Novel Monomer CDD

The obtained target products were tested by infrared spectroscopy, and the results of CDD-K infrared spectroscopy are shown in Figure 3.

As can be seen from Figure 3, there are -OH stretching vibration peaks at 3355 cm^−1^, and C=O and C=N stretching vibration peaks at 1636 cm^−1^.

According to the analysis, the infrared spectrum test results can identify the product as CDD.

An infrared spectrum test was performed on the obtained target product PCDD, and the infrared spectrum test results of the product PCDD are shown in Figure 4.

As can be seen from Figure 4, the stretching vibration peak of -NH at 3360 cm^−1^, the stretching vibration peak of C=O at 1680 cm^−1^, the ether bond characteristic peak of the C-O-C stretching vibration peak at 1256 cm^−1^, and the stretching vibration peak of Ar-Cl at 870 cm^−1^ disappear. According to the analysis, the functional groups of the target product are reflected in the infrared spectrum, and the infrared spectrum test results can determine that the product is a new monomer CDD autocondensation product PCDD.

### 3.3. Study on Ternary Copolymerization of New Monomer CDD

The obtained target product PBCD was tested by infrared spectrum, and the result of PBCD was shown in Figure 5. As can be seen from Figure 5, the -OH stretching vibration peak of BHPF at 3330 cm^−1^ disappears, the -NH stretching vibration peak of DHPZ-NH-Cl at 3368 cm^−1^ disappears, and the C=O stretching vibration peak at 1650 cm^−1^ disappears. At 1281 cm^−1^, the Ar-F stretching vibration peak of DFBP disappeared; at 1245 cm^−1^, the C-O-C stretching vibration peak of the ether bond characteristic peak appeared; at 870 cm^−1^, the Ar-Cl stretching vibration peak disappeared; at 745 cm^−1^, the cardon ring stretching vibration peak appeared. According to the analysis, the functional groups of the target product are reflected in the infrared spectrum.

Using CDD, BHPF, and DFBP as reaction monomers, ternary copolymerization was carried out in alkaline conditions with DMAC as the solvent. The obtained hydrogen NMR spectrum of PBCD is shown in Figure 6.

Figure 6 shows that in the ^1^H-NMR spectrum, the chemical shift is in the range of 6.93–6.95 ppm corresponding to position f H in the chemical structure formula of the product, in the range of 6.99–7.02 ppm corresponding to position e H, and 7.14 ppm corresponding to position d H. The chemical displacement was in the range of 7.22–7.31 ppm corresponding to c position H, 7.38–7.42 ppm corresponding to b position H, and 7.74–7.79 ppm corresponding to a position H. The hydrogen in PBCD structural unit was ascribed in the NMR spectra. The ratio of the h-peak area at the e and F positions was 1:1, indicating that the polymer structure in ternary copolymerization was the same as that corresponding to the initial feeding ratio, which proved that the ternary copolymerization of the new monomer CDD was successful. Combined with the infrared spectrum and ^1^H-NMR test results, it can be determined that the product, PBCD, was synthesized by CDD ternary copolymerization.

### 3.4. Study on Binary Copolymerization Polymerization

In the experiment, PBD, a binary copolymer of BHPF and DFBP, was firstly prepared under conventional Williamson reaction conditions. The hydrogen NMR spectrum is shown in Figure 7.

As can be seen from Figure 7, in the ^1^H-NMR spectrum, the chemical shift is at position e H in the chemical structure formula of product 6.85–6.87 ppm, at position d H in the chemical structure formula of product 7.13–7.15 ppm, and at position c H in position 7.29–7.39 ppm. The chemical shift is in the range of 7.64–7.72 ppm corresponding to b position H and 8.11 ppm corresponding to a position H. All the hydrogen in the structural formula appears in the NMR spectra.

The obtained target product was tested by infrared spectroscopy, and the results of PBD are shown in Figure 8.

Figure 8 shows that the -OH stretching vibration peak of BHPF disappears at 3338 cm^−1^, the C=O stretching vibration peak disappears at 1655 cm^−1^, the C-O-C stretching vibration peak disappears at 1245 cm^−1^, and the Ar-F stretching vibration peak of DFBP disappears at 845 cm^−1^. Furthermore, 745 cm^−1^ corresponds to the cardon ring stretching vibration peak of the single BHPF. The functional groups of the target product are reflected in the infrared spectrum, and the obtained product was determined as BHPF and DFBP binary copolymerization PBD by combining the results of the infrared spectrum test and nuclear magnetic characterization.

### 3.5. Analysis of Molecular Weight Test Results

The brown–yellow powder product PCDD, the yellow powder PBCD, and the white powder product PBD were extracted and dried by Soxhlet, and their molecular weight and molecular weight distribution were tested by gel permeation chromatography. The results are shown in Table 1. GPC spectra are shown in Appendix A

As can be seen from Table 1, the average molecular weight of PCDD obtained after self-polycondensation of the new phthalazone monomer CDD is 4.181 × 10^3^ kg/mol, the average weight molecular weight is 4.244 × 10^3^ kg/mol, the molecular weight distribution is 1.015, and the yield is 90.5%.

The average molecular weight and weight average molecular weight of PBD were 75.313 × 10^3^ kg/mol and 162.799 × 10^3^ kg/mol. The molecular weight distribution was 2.162, and the yield was 93.8%. BHPF and DFBP are monomers with high reactive activity, and high molecular weight polymers can be obtained under conventional industrial copolymerization reaction conditions. Therefore, the ternary copolymerization of new monomers with BHPF and DFBP was studied on this basis. The average molecular weight and weight of PBCD obtained after ternary copolymerization of the new phthalazone monomer CDD were 72.793 × 10^3^ kg/mol and 169.894 × 10^3^ kg/mol, respectively. The molecular weight distribution was 2.334, and the yield was 88.1%. According to the results, the new monomer CDD can undergo ternary copolymerization with BHPF and DFBP under conventional Williamson reaction conditions, and the two new monomers show good polymerization reaction activity during ternary copolymerization. The nuclear magnetic hydrogen spectrum of terpolymer shows that the proportion of groups in the backbone structure of the polymer corresponds to the proportion of monomers. According to the study, although the new monomer CDD autocondensation reaction is difficult to be carried out, the corresponding polymer can be obtained by the conventional experimental method through ternary copolymerization.

### 3.6. Study on Thermal Decomposition Temperature of Terpolymer

The thermal decomposition temperature was tested and studied by TGA under nitrogen conditions. Figure 9 shows the TGA curve of self-condensation PCDD of the new monomer CDD. It can be seen from the figure that the thermal decomposition temperature T_d__−__5%_ of 5 wt % of self-condensation PCDD of the new phthalazone monomer CDD is 309 °C, and the carbon residue rate is 40% at 500 °C. This is mainly because the introduction of the phthalazone structure in the main chain of polymer PCDD increases the free volume between polymer molecular chains, reduces the interaction between macromolecules, and improves the thermal decomposition temperature of autocondensation PCDD. However, the thermal decomposition temperature of PCDD is similar to that in reference [29], which indicates that a novel cross-linkable PAEK with pendant benzimidazole groups had been synthesized successfully. The difference is that the molecular weight of the PCDD synthesized in this study is low, which indicates that the molecular weight affects the thermal properties of PCDD, thus limiting its application.

Figure 10 shows the TGA curve of ternary copolymerization PBCD of the new monomer CDD and BHPF and DFBP binary copolymerization PBD. The TGA curve of PBD shows that the thermal decomposition temperature T_d__−__5%_ of 5 wt % of the binary copolymerization of PBD is 555 °C. Additionally, the carbon residue rate is 65% at 800 °C. The main reason is that the rigid cardon ring of fluorene increases the rigidity in the main chain of polymerization, the free volume of the polymer increases, the interaction between macromolecules is reduced, the thermal decomposition temperature is improved, and the polymer has good thermal performance.

The TGA curve of monomer CDD, BHPF, and DFBP ternary copolymer PBCD shows that the thermal decomposition temperature T_d−5%_ of 5 wt % of the polymer is 548 °C, which is slightly lower than that of binary copolymer PBD, and the carbon residue rate is 70% at 800 °C, indicating that the copolymer has good thermal properties. The introduction of nitrogen heterocyclic diazoketone groups increases the free volume of macromolecular chains, reduces the interaction between chains, improves the thermal decomposition temperature, and makes the terpolymer have good heat resistance.

This is related to the high decomposition temperature of the soluble PAEK copolymer synthesized by Zhihui Huang et al. through the nucleophilic substitution copolymerization reaction of DFBP, DFA, and BPAF [30]. Therefore, high-temperature-resistant poly(aryl ether ketone) containing nitrogen heterocyclic diazoketone group structure can be synthesized by monomer structure design and ternary copolymerization, which provides a reference value for the subsequent structural design of new monomers containing a nitrogen heterocycle structure and high-temperature resistant poly(aryl ether ketone) synthesis.

### 3.7. The Solubility of Terpolymer

The solubility of the homopolymer (PCDD), terpolymer (PBCD), and the binary copolymer PBD were tested, and the commonly used solvents CHCl_3_, THF, DMF, and DMAc were selected for testing. The solubilities of the polymers are shown in Table 2.

According to literature reports, PAEK synthesized by diphenyl ether and p-benzoyl chloride as monomers have poor solubility and can be oxidized by concentrated sulfuric acid to force them to “dissolve”, but they are insoluble in organic solvents [31]. However, the solubility of the new poly aryl ether ketones synthesized in the existing soluble poly aryl ether ketones in polar organic solvents such as dichloromethane and trichloromethane are still unsatisfactory [32], but the nitrogen-containing naphthalenone groups introduced in this study greatly improved this shortcoming.

The autocondensation and terpolymer PCDD and PBCD of CDD are soluble in low polar organic solvents such as CHCl_3_, CH_2_Cl_2_, and THF and high polar aprotic solvents such as DMF, DMAc, and DMSO at room temperature. The solubility of nitrogen-containing naphthalenone is better than that of binary copolymer PBD because of the torsion and asymmetry of the structure of nitrogen-containing naphthalenone, the introduction of the main chain of the polymer destroys the symmetry of the polymer chain segment, which is more conducive to the integration of solvent molecules into molecular chain segments, thus improving the solubility of the polymer, which is more conducive to processing and forming. Because of the excellent solubility of the synthesized PBCD, it can be processed into a film using a cast process, which greatly broadens the processing prospect of poly aryl ether ketone. The cast process is used to dissolve the raw material in an organic solvent to create a viscous solution, spreading on a flat and uniform rotating smooth support body, forming a film. This solution has the advantages of excellent uniformity and a smooth and clean surface.

## 4. Conclusions

The new phthalazone monomer CDD was selected for polymerization. The number average molecular weight of PCDD was 4.181 × 10^3^ kg/mol, the molecular weight distribution was 1.015, the number average molecular weight of PBCD was 72.793 × 10^3^ kg/mol, and the molecular weight distribution was 2.334. By comparing the properties of PBCD with BHPF and DFBP binary copolymerized PBD, the heat resistance of PBCD was slightly lower than that of PBD, but PBCD was easily dissolved in common solvents trichloromethane (CHCl_3_), tetrahydrofuran (THF), *N*,*N*-dimethylformamide (DMF), and *N*,*N*-dimethylacetamide (DMAc) at room temperature, and their solubility was better than that of PBD. This is because the introduction of diazaphanone structure provides PBCD with good heat resistance and solubility. Due to its solubility and high-temperature resistance, the processability of poly(aryl ether ketone) is greatly improved. It is expected that the solvent casting method can be used to eliminate the difficulty of forming poly(aryl ether ketone), thus providing broad prospects for its use in the miniaturization of VLSI, electronic equipment, and electronic components.

## Data Availability

Not applicable.

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
