# Peer review of "Synthesis and Characterization of a Self-Polycondensation Diazaphthalanone Monomer and Its Polymers from Polycondensation Reactions"

_polymers, 2022, doi:10.3390/polym14183904_

Round 1
Reviewer 1 Report
This research is interesting. However, the following are the comments:
1. Rewrite the abstract in the order of (a) the overall purpose of the study(b) the basic design of the study(c)major findings.
2. There is lack of research gap.
3. Make clear objectives.
4. Include percentage of purity of the chemicals used and model numbers of the equipment used by providing country names, too.
5. It seems that the synthesis part is good. However, more characterization techniques can be added like SEM, UV-ViS, etc.
6. Comparison of the findings with the literature is necessary.
7. Rewrite conclusions by including the exact findings from the results and discussion.
8. English needs to improve especially results and discussion part of the manuscript.
Author Response
Response to Reviewer 1 Comments
Dear Reviewer,
Thank you for taking time out of your busy schedule to review the manuscript. Now we have carefully corrected and replied the manuscript for this revision. The revision instructions are as follows:
Point 1: Rewrite the abstract in the order of (a) the overall purpose of the study(b) the basic design of the study(c)major findings.
Response 1: Thanks very much for the comments. According to the reviewer's suggestion, we added the overall purpose of the study and revised the purpose and main findings of the study in the abstract. Additions are highlighted in yellow on the first page of the text.
Point 2: There is lack of research gap.
Response 2: Many thanks for the insightful comments and suggestions. According to the reviewer's suggestion, we searched the recent literature on the modification of poly aryl ether ketone [31] (DOI: 10.1016/j.mtcomm.2019.100696), [32] (DOI: 10.1002/app.50895), [34] (DOI: 10.1016/j.eurpolymj.2020.110205) and compared its thermal performance and solubility, and found that the thermal performance and solubility of the nitrogen-containing heterocyclic poly aryl ether ketone studied in our study were better.
Point 3: Make clear objectives.
Response 3: Thanks very much for the comments. High performance engineering plastics polyether ketone materials have been widely used in many emerging fields of science and technology. The main chain of the polymer is mainly composed of benzene ring and ether bond, making the Polyether ketone materials with high strength, heat resistance, size stability and other excellent properties. It has been valued and applied in the fields of electronics and electrical appliances, machinery, and aerospace, etc. For example, it is mainly used in the electronic industry for wire, magnetic conductor cladding, high temperature terminal posts, terminal boards and scram wire printed circuit boards. Due to the high crystallinity, the conventional PAEKs usually have poor solubility, high melting temperature, and melt viscosity, which makes it difficult for their processing and applications in high- performance composite materials and membrane, etc. Simultaneously, the relatively low glass transition temperatures (Tg) lead to poor creep resistance that erodes the dimensional stability of polymer. Thus, it is of necessity to develop new PAEKs with high Tg and fine solubility, aiming to enhance creep resistance, provide alternative processing methods, and broaden the applicable fields.
Point 4: Include percentage of purity of the chemicals used and model numbers of the equipment used by providing country names, too.
Response 4: Thanks very much for the insightful comments. After modification, the chemical purity percentage and the name of the manufacturer are marked with a bright yellow shading in the " Materials" on the third page.
Point 5: It seems that the synthesis part is good. However, more characterization techniques can be added like SEM, UV-Vis, etc.
Response 5: Thanks a lot for your kind suggestion. We are also very interested in SEM and UV-vis characterization. However, our focus this time is specifically on the thermal properties and solubility of polymers. We hope to keep the focus unchanged, but we have considered your proposal as a consideration for future research to see if there is a better performance.
Point 6: Comparison of the findings with the literature is necessary.
Response 6: Thank you so much for your professional comments. At the suggestion of the reviewer, we added comparisons of results with recent literature. By searching the recent literature, we compared the thermal properties of the synthesized polymers with reference [31] (DOI: 10.1016/j.mtcomm.2019.100696) and reference [32] (DOI: 10.1002/app.50895) and the solubility with reference [34] (DOI: 10.1016/j.eurpolymj.2020.110205). We have highlighted the details on pages 12 to 13 with a highlighted yellow shading.
Point 7: Rewrite conclusions by including the exact findings from the results and discussion.
Response 7: Thanks very much for the insightful comments. We accepted the suggestion. We corrected the writing of the conclusions by retesting the molecular weight results, combined with the prospect of poly aryl ether ketones containing nitrogen heterocycles with good solubility and heat resistance.
Point 8: English needs to improve especially results and discussion part of the manuscript.
Response 8: Thanks very much for your comments, which are very helpful for us to improve the manuscript, and our language should be improved. After carefully check, we found many grammar and sentence errors, and have modified the manuscript accordingly. Furthermore, we have invited several English teachers help correct grammar and sentences, and we hope the revised paper will be clearer on expressions.

Reviewer 2 Report
The manuscript polymers-1898332 "Synthesis and Characterization of a self-Polycondensation diazaphthalanone monomer and Its Polymers from Polycondensation Reactions" by Liu et al. is an improved version of the manuscript polymers-1802731 and describes the synthesis of novel diazaphthalanone monomer, 2-(4-chlorophenyl)-2,3-dihydrophthalazine-1,4-dione and the study of polymerization activity. The authors have interesting experimental results, so I think that this paper will be of interest to the readers of Polymers.
The authors slightly improved the previous version of the manuscript.
Questions and comments:
1) Since many compounds (monomers and polymers) were synthesized for the first time, these compounds should be characterized by 1H, 13C NMR, IR spectroscopy and mass spectrometry. Images of all spectra should be in supplementary materials
2) About interpretation of NMR spectra. If the authors write the multiplicity of signals as a doublet, a triplet etc., then it is necessary to add spin-spin interaction constants.
3) Images of GPC spectra must be added to supplementary materials.
4) The authors need to add the information about the possible further application of the obtained results to the conclusions.
Author Response
Response to Reviewer 2 Comments
Dear Reviewer,
Thank you for taking time out of your busy schedule to review the manuscript. Now we have carefully corrected and replied the manuscript for this revision. The revision instructions are as follows:
Point 1: Since many compounds (monomers and polymers) were synthesized for the first time, these compounds should be characterized by 1H, 13C NMR, IR spectroscopy and mass spectrometry. Images of all spectra should be in supplementary materials
Response 1: Thank you for the professional comments. Your suggestions are of great help to us. I know that all the new compounds obtained need to be characterized by 1H and 13C NMR, IR spectroscopy and mass spectrometry, which will greatly improve the grade of this paper. However, we have characterized them with 1H NMR and IR spectroscopy, and these data are sufficient to justify the successful synthesis of new compounds and polymers. The characterization of 13C NMR and mass spectrometry will be reflected in further work if we have test conditions.
Point 2: About interpretation of NMR spectra. If the authors write the multiplicity of signals as a doublet, a triplet etc., then it is necessary to add spin-spin interaction constants.
Response 2: Thanks very much for the insightful comments. As suggested by the reviewer, we annotate the increased spin-spin interaction constants in red font on pages 3, 4, and 5 of the articles.
Point 3: Images of GPC spectra must be added to supplementary materials.
Response 3: Thanks very much for the comments and suggestions. We accepted this suggestion, and we remeasured the molecular weight data and added images of the GPC spectra in the supplementary information. Supplementary information is sent as a Word document named "Supplementary Information".
Point 4: The authors need to add the information about the possible further application of the obtained results to the conclusions.
Response 4: Thanks very much for the insightful comments. Through the revision of the experimental results and the solubility expression in the discussion, we have made a reasonable prospect for the novel poly aryl ether ketone in the conclusion. Because of its good solubility, it can be poured into film by organic solvent, and has a broad application prospect in the field of electronic device film.
The annotated manuscript will be uploaded along with the review comments, and I will upload an unannotated manuscript at last

Reviewer 3 Report
The paper devoted to the synthesis of the Polyether ketone (PEK) plastics synthesis and characterization is well-written and clearly organized. The results are novel. The theme is actual.The improvement of solubility of PEEK is an important problem.
Remarks:
The paper does not supply the results of the statistical treatment of the reported results. This is a very bad methodological mistake.
1. The statistical scattering of the results supplied in Table 1 should be reported in the revised version of the manuscript.
2. In the text:
"The number average molecular weight of PCDD was 4.181 × 103, the molecular weight distribution is 1.015, and the number average molecular weight of PBCD is 72.793 × 103, the molecular weight distribution is 2.334."
The statistical scattering of the reported results should be supplied.
3. It should be mentioned that the improvement of solubility of PEEK decreases its chemical stability; this is inevitable.
Author Response
Response to Reviewer 3 Comments
Dear Reviewer,
Thank you for taking time out of your busy schedule to review the manuscript. Now we have carefully corrected and replied the manuscript for this revision. The revision instructions are as follows:
Point 1: The statistical scattering of the results supplied in Table 1 should be reported in the revised version of the manuscript.
Response 1: Thank you for the professional comments. The molecular weight data and statistical chart of molecular weight distribution data in Table 1 have been reflected in the supplementary information, which has been uploaded to the website
Point 2: In the text:
"The number average molecular weight of PCDD was 4.181 × 103, the molecular weight distribution is 1.015, and the number average molecular weight of PBCD is 72.793 × 103, the molecular weight distribution is 2.334."
The statistical scattering of the reported results should be supplied.
Response 2: Thanks very much for the insightful comments. We accepted this suggestion, and we remeasured the molecular weight data and added images of the GPC spectra in the supplementary information. Supplementary information is sent as a Word document named "Supplementary Information".
Point 3: It should be mentioned that the improvement of solubility of PEEK decreases its chemical stability; this is inevitable.
Response 3: Thanks very much for the insightful comments. As you said, there is a negative relationship between solubility and heat resistance to a certain extent. The aim of the polymers we studied was to develop a novel poly aryl ether ketone with high temperature resistance and solubility, thus introducing a structure called 2 - (4-chlorophenyl) -2, 3-Dihydrophthalazine-1, 4-Dione (CDD). This gives the poly aryl ether ketones distorted non-coplanar properties. the twisted noncoplanar structure leads to substantial entanglement in molecular chains. The regular accumulation of polymer chains is interrupted, which endows the PAEK with excellent solubility in common solvents. On Other hand, the mobility of molecular segments is still hindered and lots of entanglement of molecular chains exist, enhancing the thermal resistance and mechanical performance of PAEK.
Reviewer 4 Report
The paper deals with the synthesis and characterization of polycondensation di-azaphthalanone monomers and related derived polymers. The paper is well organized as regards experimental part and the test results are presented in a concise and clear mode. The authors should explain for what technological procedures a higher solubility of polyether ketone is needed, and how the newly obtained polymer derivates respond to such a need. On the other hand, the conclusions must be expanded in order to better synthesize the experimental innovation and the relevant results in terms of physical properties. On the other hand, the difficulty of poly (aryl ether ketone) 'molding' is mentioned, and such assumptions should be related to the technological procedures that need a higher solubility of polyether ketone, as mentioned above, in order to improve the practical addressability of the study.
Author Response
Response to Reviewer 4 Comments
Dear Reviewer,
Thank you for taking time out of your busy schedule to review the manuscript. Now we have carefully corrected and replied the manuscript for this revision. The revision instructions are as follows:
Point: The paper deals with the synthesis and characterization of polycondensation diazaphthalanone monomers and related derived polymers. The paper is well organized as regards experimental part and the test results are presented in a concise and clear mode. The authors should explain for what technological procedures a higher solubility of polyether ketone is needed, and how the newly obtained polymer derivates respond to such a need. On the other hand, the conclusions must be expanded in order to better synthesize the experimental innovation and the relevant results in terms of physical properties. On the other hand, the difficulty of poly (aryl ether ketone) 'molding' is mentioned, and such assumptions should be related to the technological procedures that need a higher solubility of polyether ketone, as mentioned above, in order to improve the practical addressability of the study.
Response: Thanks very much for the insightful comments. We have added to the discussion of solubility a practical method: cast process. This is the main method for mass production of thin films, usually by dissolving the polymer with an organic solvent and then volatilizing the solvent to form the polymer film. The details are highlighted in bright yellow shading on page 15 in the last paragraph of results and discussion. And considering that discussion, the outlook in the conclusion becomes more concrete. Changes are also highlighted with a bright yellow shading.
The annotated manuscript will be uploaded along with the review comments, and I will upload an unannotated manuscript at last

Round 2
Reviewer 1 Report
The revised manuscript looks ok.
Reviewer 2 Report
I thank the authors for answering my questions and improving the manuscript.